

# The effect of the present-day imbalance on schematic and climate forced simulations of the West Antarctic Ice Sheet collapse

Tim van den Akker[1], William H. Lipscomb[2], Gunter R. Leguy[2], Willem Jan van de Berg[1], Roderik S.W. van de Wal[1,3,4]

[1]Institute for Marine and Atmospheric Research Utrecht, Utrecht University, Netherlands

[2]Climate and Global Dynamics Laboratory, NSF National Center for Atmospheric Research, Boulder, CO, USA

[3]Department of Physical Geography, Utrecht University, Netherlands

[4]KNMI Royal Netherlands Meteorological Institute, De Bilt, Netherlands

*Correspondence to*: Tim van den Akker (t.vandenakker@uu.nl)

## Abstract

Recent observations reveal that the West Antarctic Ice Sheet is rapidly thinning, particularly at its two largest outlet glaciers, Pine Island Glacier and Thwaites Glacier, while East Antarctica remains relatively stable. Projections give a mixed picture, some model project mass gain by increased surface mass balance, most models project some or severe mass loss by increasing

ice discharge. In this study, we explore the effect of present-day ice thickness change rates on forced future simulations of the Antarctic Ice Sheet using the Community Ice Sheet Model (CISM). We start with a series of schematic, uniform ocean temperature perturbations to probe the sensitivity of the modelled present-day imbalance to ocean warming. We then apply ocean and atmospheric forcing from seven ESMs from the CMIP5 and CMIP6 ensemble to simulate the Antarctic Ice Sheet from 2015 to 2500. The schematic experiments suggest the presence of an ice-dynamical limit, TG cannot collapse before

~2100 without more than 2 degrees of schematic, suddcen and uniform ocean warming. Meanwhile, the maximum GMSL rise rate during the collapse increases linearly with ocean temperature, indicating that while earlier collapse timing shows diminishing returns, the rate of sea-level rise keeps on intensifying with stronger forcing. In the simulations driven with ESM forcing, including or excluding the present-day imbalance contributes for the West Antarctic Ice Sheet as much to the uncertainty in the mass loss rates in the coming 5 centuries as the choice of ESM forcing. For the East Antarctic Ice Sheet on

shorter timescales (until 2100), adding the present-day observed mass change rates doubles its global mean sea level rise contribution. On longer timescales (2100–2500), the effect of the present-day observed mass change rates is smaller. Thinning of the West Antarctic Ice Sheet induced by the present-day imbalance is to a small degree partly compensated by present-day ice sheet thickening of the East Antarctic Ice Sheet over the coming centuries, which persists in our simulations. Moreover, these deviations are overshadowed by the mass losses induced by the projected ocean warming. The relative importance of

including the observed present-day mass loss rates decreases for larger (ocean) warming under climate forcing, and decreases over time.



# 1 Introduction

The lastest IPCC estimate of the Antarctic Ice Sheet (AIS) contribution to global mean sea level (GMSL) rise ranges from 0.03 m (SSP1-1.9, low end of the likely range) to 0.34 m (SSP5-8.5, high end of the likely range) in 2100 (Fox-Kemper et al.,
2021). This is an assessment based on ice sheet models, which simulate the future behavior of the AIS. The uncertainty in ice sheet modelled sea level rise from the AIS until 2100 is relatively low because the major floating ice shelves keep the grounded ice sheet currently in place (Van De Wal et al., 2022), but increases after 2100 because non-linear processes could accelerate mass loss significantly (Fox-Kemper et al., 2021; Payne et al., 2021; Deconto et al., 2021). It is argued that the main contributors to Antarctic mass loss uncertainty on the long term (e.g. after 2100) are the choice of ice flow model and the
choice of the Earth System Model (ESM) used as ocean and atmospheric forcing (Pattyn and Morlighem, 2020; Aschwanden et al., 2021).

Both ice flow models and ESMs have grown in number over the past decades. This has increased the capability to quantify the uncertainty related to the choice of the model and to the choice of ESM forcing. The Ice Sheet Model Intercomparison for CMIP6 (ISMIP6, Nowicki et al. (2016); (2020)) exemplifies multi-model ensemble simulations of the AIS and state-of-the-
art quantification of different sources of uncertainty in sea level rise projections. Seroussi et al. (2023) show that until 2100, the choice of ice sheet model (which encompasses all modellers choices made like the momentum balance approximation, resolution, initialization method and parameterizations) is the dominant source of uncertainty of projected GMSL from the AIS, with a growing uncertainty caused by the choice of ESM forcing over time. This study shows large geographic differences: for example, the variance associated with the ice sheet model is large for Thwaites Glacier (TG) and Pine Island
Glacier (PIG), and small for the MacAyeal and Whillans glaciers. In a follow-up study, Seroussi et al. (2024) show that the choice of ice flow model remains the largest source of uncertainty until 2300, with ESM forcing as second largest contributor. A prominent difference between ice flow models is their method used to simulate the present-day state of the AIS, here referred to as the initialization method.

To obtain a good model representation of the present-day configuration of the AIS it is necessary to do a model inititialization.
In an initialization, modellers often include the observed ice thickness and/or the observed ice surface velocities as target variables for the model to match. Modellers may also include present-day mass change rates as a target variable during initializations, as in Data-Assimilation methods (e.g. Larour et al. (2012); Gudmundsson et al. (2012); Bradley and Arthern (2021)), In these methods, the ice sheet model adjusts model parameters by comparing modelled and observed ice surface velocities while imposing observed ice thickness. This process can lead to nonzero ice thickness change rates, or drift, when
the model is run forward in time, even without external climate forcing. The model drift can be largely removed by including it in the surface mass balance (SMB) (Bett et al., 2023), or the drift can be nudged towards observed mass change rates (Rosier et al., 2024). All three studies by Rosier et al. (2021), Bett et al. (2023) and Rosier et al. (2024) state that it is impossible to



obtain a perfect fit between observed and modelled ice thickness, ice surface velocities, and mass change rates simultaneously because the three datasets are not mutually consistent.


The other often-used ice sheet model initialization method is the so-called 'spin-up' method (Winkelmann et al., 2011; Pollard and Deconto, 2012; Greve and Blatter, 2016; Quiquet et al., 2018; Lipscomb et al., 2019; Berends et al., 2021; Berends et al., 2022). For this method a long simulation towards present-day, or a simulation freely evolving from the present-day ice thickness distribution can be used. Optionally, uncertain parameters can be tuned to decrease the misfit between observed and

modelled ice thickness and/or ice surface velocities, during the simulation. This results in a modelled ice sheet close to the target observables (ice thickness and/or ice surface velocities) and in steady state. To get the modelled ice sheet to exhibit a mass change rate, ideally also close to observations, a historical forcing scenario can be used (Reese et al., 2023; Coulon et al., 2024; Klose et al., 2024). Recently, Van Den Akker et al. (2025b) developed a method to incorporate the observed mass change rates at grid-point base in this spin-up initialization, in such a way that the resulting modelled ice sheet can start future

simulations immediately from the observed imbalance. This circumvents the need for a historical simulation and forcing over the near-present period. Van Den Akker et al. (2025b) show that initialization with the present-day observed mass change rates will always lead to an unforced collapse (i.e., without additional ocean or atmospheric warming) of the West Antarctic Ice Sheet (WAIS), starting with Thwaites Glacier (TG) and Pine Island Glacier (PIG). The rapid collapse phase typically begins after a period of 500 - 2000 years of slow retreat. However, that study did not investigate how forced future evolution scenarios

are affected by including the present-day mass change rates.

In this study, we focus on forced simulations of the Antarctic Ice Sheet, using firstly schematic ocean warming, and secondly ocean temperature and SMB anomalies from ESMs from the CMIP5 and CMIP6 ensembles following either the SSP1-2.6, SSP5-8.5, RCP1-26 or the RCP 5-85 scenario. The schematic forcing consitst of a targeted (e.g. only at TG and PIG) sudden

uniform ocean warming up to 2 degrees. The ESM forcing used in this study has also been used as forcing for the ISMIP6 Antarctic Ice Sheet study in Seroussi et al. (2024). These anomalies serve as input to long-term future simulations from 2015 to 2500, to capture the longer-term effects of $21^{st}$ century climate change on the mass of the ice sheet. We use two initializations of the Antarctic ice sheet to start our future simulations, namely with and without the observed mass change rates. Hence, the latter starts from steady-state. With these simulations we will quantify the importance of the current imbalance of the Antarctic

Ice sheet compared to projected future changes in ocean temperature and SMB. Our null hypothesis is that the GMSL rise from the present-day mass loss rates is independent of the GMSL rise caused by an increase in ocean thermal forcing, i.e. that the present-day mass loss rates do not influence future forced projections. In section 2, we introduce the Community Ice Sheet Model, and we discuss the general initialization procedure and the oceanic and atmospheric ESM forcings used. In section 3, we show the results of the forced simulations. Section 4 contains the discussion, followed by conclusions in section 5.



## 2 Methods

### 2.1 Community Ice Sheet Model (CISM)

The Community Ice Sheet Model (CISM, (Lipscomb et al., 2019; Lipscomb et al., 2021)) is a thermo-mechanical higher-order ice sheet model, which is part of the Community Earth System Model version 2 (CESM2 (Danabasoglu et al., 2020)). Earlier applications of CISM to Antarctic Ice Sheet retreat can be found in Lipscomb et al. (2021); Berdahl et al. (2023); Van Den Akker et al. (2025b); Van Den Akker et al. (2025a). We run CISM on an equidistantial 4 km grid. The variables and constants used in the text and equations below are listed in Supplementary Tables S1 and S2.

We run CISM with a vertically integrated higher-order approximation to the momentum balance, the Depth Integrated Viscosity Approximation (DIVA) (Goldberg, 2011; Lipscomb et al., 2019; Robinson et al., 2022):

$$\frac{\partial}{\partial x}\left(2\bar{\eta}H\left(2\frac{\partial \bar{u}}{\partial x}+\frac{\partial \bar{v}}{\partial y}\right)\right)+\frac{\partial}{\partial y}\left(\bar{\eta}H\left(\frac{\partial \bar{u}}{\partial y}+\frac{\partial \bar{v}}{\partial x}\right)\right)-\beta u_{x,b}=\rho_i gH\frac{\partial s}{\partial x} \tag{1.1}$$

$$\frac{\partial}{\partial x}\left(\bar{\eta}H\left(\frac{\partial \bar{u}}{\partial y}+\frac{\partial \bar{v}}{\partial x}\right)\right)+\frac{\partial}{\partial y}\left(2\bar{\eta}H\left(2\frac{\partial \bar{v}}{\partial v}+\frac{\partial \bar{u}}{\partial x}\right)\right)-\beta u_{y,b}=\rho_i gH\frac{\partial s}{\partial y} \tag{1.2}$$

In which $\bar{\eta}$ is the depth-averaged viscosity, $H$ the ice thickness, $\bar{u}$ and $\bar{v}$ respectively the depth-averaged ice velocities in the x- and y-direction, $\rho_i$ the density of glacial ice and $s$ the surface height above sea level. Basal friction, which appears as the product of a non-negative scalar $\beta$ and the directional velocity in Eqs. 1.1 and 1.2, can be parameterized in several ways. In this study we use the regularized Coulomb sliding law suggested by Zoet and Iverson (2020):

$$\tau_b = \beta u_b = C_c N\left(\frac{u_b}{u_b+u_0}\right)^{\frac{1}{m}} \tag{1.3}$$

where $C_c$ is a unitless parameter in the range [0,1] controlling the strength of the regularized Coulomb sliding, $u_b$ the ice basal velocities and $u_0$ and $m$ are free parameters. The effective pressure $N$ is estimated according to Leguy et al. (2014) and Leguy et al. (2021), assuming full ocean connectivity and based on the height above floatation of the grounded ice. This implies that the scheme proposed by Leguy et al. (2014) accounts for basal water pressure (which reduces $N$) only near grounding lines and not in other parts of the ice sheet; thus the effective pressure equals ice overburden for most of the ice sheet.

Since $C_c$ is poorly constrained by theoretical considerations and observations, we use it as a spatially variable tuning parameter. We tune the logarithm of $C_c$ using a nudging method (Lipscomb et al., 2021; Pollard and Deconto, 2012):





$$\frac{dC_l}{dt} = -\left(\frac{H - H_{\text{obs}}}{H_0\tau}\right) - \frac{2}{H_0}\frac{dH}{dt} - \frac{r}{\tau}(C_l - C_{lr}) + \frac{L^2}{\tau}\Delta C_l \tag{1.4a}$$

$$C_l = \log_{10} C_c \tag{1.4b}$$

The logarithmic relaxation target $C_{lr}$ is a 2D field that penalizes very high and low values of $C_c$. It is based on elevation, with lower values at low elevation where soft marine sediments are likely more prevalent, following Winkelmann et al. (2011). We chose targets of 0.1 for bedrock below -700 m asl and 0.4 for 700 m asl, with linearly interpolation in between, based on Aschwanden et al. (2013). The motivation for this is that lower elevations deglaciate earlier and therefore have more softer marine till relative to higher elevated bedrock, since they likely were deglaciated more in the past compared to regions with

higher bedrock. The last term on the RHS of Eq. (1.4) is new compared to Lipscomb et al. (2021), Van Den Akker et al. (2025a) and Van Den Akker et al. (2025b) and is introduced to smooth the pattern of inverted $C_c$ by surpressing large spatial gradients. Additionally, the smoothing reduces the model drift. Choosing the parameters $(H_0, \tau, r, L)$ regulates the size and therefore the importance of the different terms to each other. Their values are presented in Table S2.

Basal melt rates *(BMR)* are calculated using a quadratic relation with a sub-shelf thermal forcing observational dataset of Jourdain et al. (2020) during the initialization and the forced simulations:

$$BMR = \gamma_0 \left(\frac{\rho_w c_{pw}}{\rho_i L_f}\right)^2 (\max[TF_{\text{base}} + \delta T, 0])^2 \tag{1.5}$$

in which the baseline thermal forcing, $TF_{\text{base}}$, is the difference between the melting point and the ocean temperature at the base of the modelled ice shelf, and $\delta T$ a local correction temperature. This basal melt parameterization was developed and tested by Jourdain et al. (2020) and Favier et al. (2019) with the purpose of modelling present-day Antarctic basal melt rates.

The quadratic relationship between thermal forcing and basal melt rates reflects a positive feedback. As the ice shelf melts, freshwater is added to the cavity, which is more buoyant than the saltier ocean water. This causes the sub-shelf meltwater plume to rise faster, and through erosion and upwelling of new warm ocean water, the basal melt rates will increase. Eq. (1.5) parameterizes this feedback to a reasonable approximation for present-day basal melt rates, but it is unknown how well it simulates basal melt rates for several degrees of (future) ocean warming.


The basal melt rates are tuned through the local correction temperature $\delta T$ such that the floating ice matches as closely as possible the thickness observations of Morlighem et al. (2020) following a similar procedure as for friction (Eq. 1.4):

$$\frac{d(\delta T)}{dt} = T_s\left[\left(\frac{H - H_{\text{obs}}}{H_0\tau}\right) + \frac{2}{H_0}\frac{dH}{dt}\right] + \frac{(T_r - \delta T)}{\tau} + \frac{L^2}{\tau}\Delta\delta T, \tag{1.6}$$





in which $T_s$ is the temperature scale of the inversion (0.5 K in this study). This tuning increases (decreases) the local correction temperature $\delta T$ in grid cells where the modelled ice shelf is too thick (thin), increasing (decreasing) the basal melt rates. The tuning includes a relaxation target $T_r$, being 0, to penalize large deviations from the dataset of Jourdain et al. (2020). The melt sensitivity $\gamma_0$ is chosen to be $3.0 \times 10^4$ m/yr, which was used in Lipscomb et al. (2021); Van Den Akker et al. (2025b); Van Den Akker et al. (2025a) to obtain basal melt rates in agreement with observations and a shelf average $\delta T$ close to zero in the

Amundsen Sea Embayment, where currently the largest ice shelf melt rates are observed (Adusumilli et al., 2020). The last term is added to, just as for the $C_c$ inversion in Eq. (1.4), surpress large spatial variations in the inverted ocean temperature perturbation field, as large spatial gradients of several degrees between grid cells is physically implausible.

To prevent abrupt jumps in modeled quantities at the grounding line (such as basal friction and basal melt), we use a

grounding line parameterization from Leguy et al. (2021):

$$f_{\text{float}} = -b - \frac{\rho_i}{\rho_w} H \tag{1.7}$$

which varies smoothly from negative for grounded ice to positive for floating ice. The variable $f_{\text{float}}$ is used to compute the floating fraction as a percentage of grid cell area by bilinearly interpolating its value from cell vertices to the cell areas scaled to the cavity thickness (Leguy et al. (2021). The grounded and floating fraction of a cell are then used to scale basal friction and basal melting. For the basal melting, this is referred to as the Partial Melt Parameterization (PMP), see Leguy et al. (2021).


There are several calving laws in the literature e.g. Yu et al. (2019); Wilner et al. (2023); Greene et al. (2022). However, there is no agreed-upon best approach to Antarctic calving, and most calving laws struggle to reproduce the observed calving front at multiple locations simultaneously without adjusting local parameters (Amaral et al., 2020). We therefore choose to apply a simple no-advance calving scheme, preventing the calving front from advancing beyond the observed present-day location.

The ice shelf front can retreat, but only when the ice thins below a threshold thickness of 1 m. This implies that the ice shelf front can only retreat when the basal melt rates are sufficiently high to remove floating ice upstream of the present-day calving front. This is a conservative approach, ignoring the possibility of calving-front retreat through shelf thinning or of shelf collapse by hydrofracturing. The implementation of a more physically-based calving law in CISM is the topic of ongoing research.

**2.2 Initializations**

We perform initializations with or without incorporating the observed present-day mass change rates, hence a transient and equilibrium initialization, respectively. For the equilibrium initialization, $C_c$ and $\delta T$ are tuned using Eqs. (1.4) and (1.6) until the modelled ice sheet is in equilibrium, thus $\partial H/\partial t = 0$, given by

$$\frac{\partial H}{\partial t} = -\nabla F + B \tag{1.8}$$



In Eq. (1.8), $\nabla F$ is the ice flux divergence, and $B$ the sum of the basal and SMBs. For the transient initialization, $C_c$ and $\delta T$ are

tuned using mass conservation complemented by the present-day observed mass change rates as was done by Van Den Akker

et al. (2025b), using

$$\frac{\partial H}{\partial t} = -\nabla F + B - \left.\frac{\partial H}{\partial t}\right|_{obs} \tag{1.9}$$

in which the last term, the pseudo-flux, is the observed mass change from Smith et al. (2020). By subtracting this observed

mass change, mass is added during the initialization where thinning is observed. After the initialization and for normal forward

simulations, this pseudo-flux is removed, so that model simulations start with an imbalance and a thinning rate closely

matching the observed thinning rates. Both initializations are run for 10 kyr, which proved to be long enough to reach an

equilibrium. We evaluate both initializations by comparing to observations of ice thickness (Morlighem et al., 2020), ice

surface velocities (Rignot et al., 2011), total basal melt fluxes (Adusumilli et al., 2020; Rignot et al., 2013) and by evaluating

their model drift. We test the drift by running forward for 1000 years (i.e., to the year 3015) without forcing, with the inverted

parameters (e.g. $\delta T$ and $C_c$ in Eqs. (1.4) and (1.6) kept constant, and with continuing adding the observed mass changes for

the transiently initiated model state as described by Eq. (1.9). This is a time scale longer than our period of interest, which runs

only to 2500, thus for 485 years. The resulting drift, shown in Figures S1 and S2, results in a change of about 0.04% in the ice

volume above floatation over the course of 1000 years. Changes in ice sheet thickness are small. These results ensure that we

can attribute any major changes in ice sheet mass to the applied forcing and not to model drift.

## 2.3 Continuation simulations

We first test the sensitivity of the two initializations to schematic and sudden ocean warming by conducting 11 idealized

experiments with sudden and sustained warming, only in the ASE region shown in Figure S3. We raise the ocean temperatures

in these schematic tests, which appears as the sum $TF_{\text{base}} + \delta T$ in Eq. (1.5), from 0 to 2 K with steps of 0.2 K.

We then perform two sets (i.e. starting from the transient and starting from the equilibrium initialization) of simulations for

each of the seven ESMs, introduced in the next section. We timestamp the end of our initialization at 2015 based on the

observational datasets used to calibrate the model (Smith et al., 2020; Morlighem et al., 2020; Rignot et al., 2011). We then

run CISM forward for 485 years to 2500. The seven forcing datasets from five different ESMs span the period 1995–2100,

with four ESMs continuing until 2300 and one where the forcing was repeated from 2100 onwards. For the last 200 years of

the simulations, we fix the thermal forcing and SMB anomalies at the last datapoint year being 2300. We consider the whole

continent in our analysis, but focus on areas with large changes and potentially large sea level contributions, like the Amundsen

Sea Embayment, the Filchner-Ronne basin, and the Ross basin. Those areas are shown in Figure S3.



## 2.4 Climate Forcing

We use the same set of ocean and SMB forcings as Seroussi et al. (2024). The ocean and SMB forcings stem from seven ESM simulations from the CMIP5 and CMIP6 ensemble. Two models from the CMIP5 ensemble were selected by Barthel et al. (2020): the Community Climate Model (CCSM4) and the Hadley Centre Global Environment Model (HadGEM2-ES). Additionally, two CMIP6 participating models are used: the Community Earth System Model (CESM2) and the UK Earth System Model (UKESM). The Norwegian Earth System Model (NorESM) was used as a reference run in the ISMIP6 ensemble and will be used in this study as well. The models HADGEM2-ES, CESM2 and UKESM have a climate sensitivity to doubling of $CO_2$ concentrations at the upper end of the 90% confidence interval in the IPCC-AR6 report (Meehl et al., 2020). More information on the selection of these ESM forcings can be found in Barthel et al. (2020) and Seroussi et al. (2024).

**Figure 1. Thermal forcing ($TF_{base} + \delta T$ in Eq. (1.5)) from the seven ESMs used.** Thermal forcing averages for the years 2280 – 2300 (except the spinup) are shown for a depth of -500 metres. Cells with bedrock above sea level are shown in grey.



The ocean forcings from the ESMs are applied as the thermal forcing $TF_{\text{base}}$ in Eq. (1.5). For this set of simulations, cavity-
resolving ocean models were not available among the CMIP5 and CMIP6 ensembles. Therefore, the ocean thermal forcing
from the ESMs is interpolated into the ice shelf cavities by Jourdain et al. (2020). First, a marine connection mask is generated,
marking cells with subzero topographic paths to the open ocean. Next, empty cells adjacent to filled ones and connected to the
ocean are identified, following Jourdain et al. (2020). These cells are then filled with the average of neighboring filled values.
Any empty cells below the new fill may remain if the local bedrock is deeper than in adjacent columns. In those cases, the
thermal forcing is linearly extrapolated using a depth dependent freezing point correction with a temperature coefficient of
7.64 x $10^{-4}$ K m$^{-1}$ from Beckmann and Goosse (2003). The average of the last 20 years (2280 – 2300) of all 7 oceanic forcings,
including the extrapolation below present-day grounded ice, relative to $TF_{\text{base}}$, are shown in Figure 1.

The computed ocean thermal forcing dataset has 30 layers with a vertical resolution of 60 metres and extends down to 1770 m
below sea level. The thermal forcing at the ice-shelf base is linearly interpolated between vertically adjacent ocean layers. If
the ice lies below the lowest ocean layer, which can occur in deep troughs beneath the Filchner-Ronne and Amery shelves, the
thermal forcing is extrapolated from the lowest layer using the depth-dependent freezing point temperature coefficient from
Beckmann and Goosse (2003).

Figure 2 shows the SMB anomalies of the seven ESMs. Large differences in magnitude are visible, with NorESM_RCP26 and
NorESM_RCP85 having hardly any change in SMB compared to present day. On the other hand, CESM2_SSP585 and
UKESM_SSP585 show large areas where the SMB decreases considerably due to surface melt. For many of these areas the
local SMB becomes net negative, which does occur at present at very few locations only (Mottram et al., 2021; Van Wessem
et al., 2018). The simulations that show considerable SMB reduction by surface melt over the ice shelves (CCSM4_RCP85,
CESM2_SSP585, HADGEM_RCP85 and UKESM_SSP585) also project an increasing SMB in the course of time for the
WAIS ice divide and in Dronning Maud Land. The latter location currently shows thickening as well (Smith et al., 2020).



**Figure 2. Surface mass balance (SMB) anomalies simulated by the seven ESMs.** The average of the years 2280 – 2300 is shown.
Anomalies are added directly to the modelled SMB annually in the continuation simulations. The observed grounding line position is shown
in black.

## 3. Results

### 3.1 Initialization evaluation

Table 1 presents the key performance metrics for the equilibrium and transient initializations, namely ice thickness, surface
ice velocities and grounding line position biases. Supplementary Figures S4 and S5 show the spatial patterns of the ice
thickness and surface velocity errors with respect to observations, and the inverted quantities for both initializations.

Overall ice thickness biases are low (Table 1), especially compared to the velocity errors. In both initializations, we tune
towards an observed ice thickness target and not to an ice surface velocity target. The velocity biases are relatively high, but
still of the same order of magnitude as many ISMIP6 models (see Seroussi et al. (2020) and Goelzer et al. (2020)). Including



the present-day mass change rates increases the ice thickness and ice velocity RMSE slightly. Furthermore, in areas where large thinning rates are observed, such as PIG and TG, the dynamic imbalance pseudo-flux in Eq. (1.8) adds considerable mass during the spin-up, which brings the modelled ice fluxes across the grounding line in the transient initialization more in line
with observations than in the equilibrium initialization, similarly to as discussed by Van Den Akker et al. (2025b); Van Den Akker et al. (2025a).

In areas with observed thickening in the dataset of Smith et al. (2020), such as at the EAIS, the sum of all ice fluxes including the pseudo-flux in Eq. (1.9) can become of similar or larger magnitude as the negative of the SMB. This requires for these
locations, for an equilibrated transient ice sheet, a largely reduced ice flux divergence or even ice flux convergence compared to the equilibrated equilibrium ice sheet. Since ice velocities are low at these locations, and therefore the ice flux is small, the basal friction cannot always decrease the ice velocity enough to reach a steady state with the correct ice thickness, and hence the ice sheet thins locally until an new equilibrium is reached. Consequently, adding the observed present-day mass change rates in these locations yields a negative modelled ice thickness bias, visible on the EAIS in Figure S2. This increases the
RMSE of the modelled ice thickness with respect to observations.

The ice surface velocity biases are generally low in both initializations, except for the Siple coast glaciers and the Ronne ice shelf. Regarding the Siple Coast, the thickness biases are low, indicating that the friction inversion in these spots can nudge the modelled ice thickness close to observations, typically with a low $C_c$ in the ice streams. This apparently leads to an
overestimation of the ice surface velocity in these ice streams, which can be counteracted by locally tuning parameters related directly to the ice velocities, like the viscosity or the flow enhancement factor. However, the relative error in these ice streams, where the ice surface velocities exceed 2 km yr$^{-1}$, is still small.

The same holds for the Ronne shelf, where the modelled ice surface velocities are too low. The ice thickness error at the
Filchner-Ronne shelf in both initializations is low, again indicating that the nudging of $\delta T$ is capable in reproducing the observed ice thicknesses. However, the nudging of $\delta T$ does not alter the modelled ice thickness by changing the ice velocities as it does for the friction inversion. To obtain a better fit, just as with the glaciers at the Siple Coast, parameters directly related to the ice velocities can be nudged. This decreases the misfit between modelled and observed ice surface velocities as shown by Van Den Akker et al. (2025a), but it also introduces another time-constant but spatially varying parameter with associated
uncertainties and the risk of over-tuning.




**Table 1. Key metrics of performance for the four initializations used in this study.** RMSE's are calculated using grid cells that contain both an observational and modelled value of the to be considered variable.

| | Equilibrium Initialization | Transient Initialization |
|---|---|---|
| **RMSE thickness [m]** | 30 | 35 |
| -      RMSE thickness grounded ice | 23 | 31 |
| -      RMSE thickness floating ice | 50 | 44 |
| **RMSE velocity [m/yr]** | 130 | 143 |
| -      RMSE velocity grounded ice | 98 | 112 |
| -      RMSE velocity floating ice | 202 | 201 |
| **RMSE grounding line position [km]** | 1.51 | 1.49 |

## 3.2 Schematicly forced Antarctic Mass change

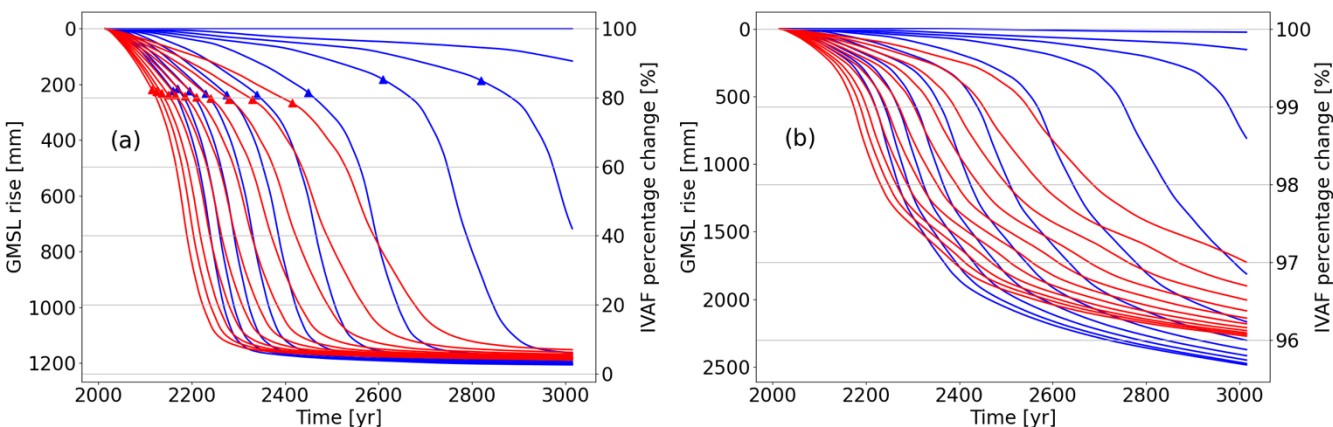


**Figure 3. Integrated ice mass response to sudden, uniform ocean warming of the ASE sector. (**left) integrated mass loss in the ASE, where Thwaites Glacier and Pine Island Glacier are situated. Left y-axis shows the GMSL rise in mm and the Ice Volume Above Floatation (IVAF) in percentage of the initial value. On the right, the same for the whole AIS. Red lines indicate simulations starting with the observed mass change rates, blue lines indicate simulations starting from an equilibrium. From right to left (later collapse to earlier collapse) the

simulations are forced with 0 to 2 K of ocean warming with steps of 0.2 K. Triangles indicate the timesteps when the line AB in Fig S6 ungrounds completely.

Figure 3 presents the integrated GMSL rise resulting from the schematic warming experiments, for both the ASE and the broader AIS. Starting from the transient initialization, applying a uniform and abrupt ocean warming to the ASE triggers an

earlier onset of collapse, with diminishing sensitivity at higher ocean temperatures, but a larger rate of GMSL rise which





increases linearly with additional ocean warming. The impact of ocean warming is more pronounced in simulations initialized without present-day mass change rates. This can be attributed to the inverted parameter $\delta T$, which tends to be lower in these runs. As a result, adding 0.2 K of ocean warming leads to a relatively larger increase in effective ocean temperatures compared to simulations that include present-day mass change rates, where $\delta T$ is typically higher.


Interestingly, the equilibrium simulations (blue lines in Fig. 3) show a greater long-term GMSL contribution from the AIS compared to the transient simulations. The equilibrium runs are more sensitive to ocean warming, which can be partly attributed to a larger relative increase in $\delta T$, and to the stabilizing effect of including present-day mass change rates. Specifically, incorporating these rates makes the Kamb Ice Stream (see Fig S6). more stable, as it is currently thickening

(Smith et al., 2020). A more stable Kamb ice stream acts as a brake on the retreat of Siple Coast glaciers once the ASE has lost its grounded ice. In this scenario, the retreat of the grounding line and subsequent ice sheet collapse are less able to propagate beyond the ASE, across the WAIS ice divide, and into the Siple Coast region. In contrast, in the equilibrium simulations, where this stabilizing effect is absent, such large-scale collapse can occur more easily.

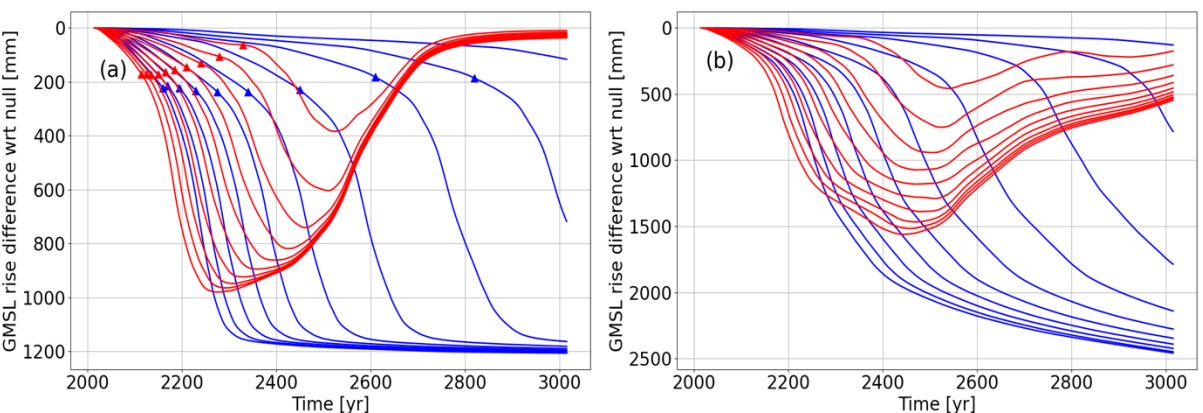


**Figure 4. All schematic warming scenarios with the control experiment (i.e. 0 ocean warming) substracted**. For the ASE (left) and the AIS (right) in terms of GMSL rise. Triangles indicate the onset of the modelled WAIS collapse. Red lines indicate simulations starting with the observed mass change rates, blue lines indicate simulations starting from an equilibrium. From right to left (later collapse to earlier collapse) the simulations are forced with 0.2 to 2 K of ocean warming with steps of 0.2 K. Triangles indicate the timesteps when the line AB

in Fig S6 ungrounds completely.

To test our null hypothesis that present-day mass loss rates do not influence future forced projections, we subtracted the results of the no-perturbation simulations (i.e. 0 K of added ocean warming in the ASE) from their corresponding schematically forced simulations. These results are shown in Figure 4. In the case of the equilibrium initialization, the no-perturbation experiment resulted in negligible mass loss, so the mass loss in the forced experiments due to the applied ocean forcing is almost the same

as the difference between the forced scenarios and the no-perturbation experiment. In contrast, the unforced transient



initialization exhibits already a collapse of the WAIS, so Figure 4 shows the deviation from this evolution due to the added ocean forcing.

Except for the first 50 years, the red and blue lines never align, implying that the present-day mass loss rates and a schematic forcing to the ASE sector does not add up linearly, contradicting our null hypothesis. For the first 250 (strong forcing) to 600
(weak forcing) years, the impact of ocean warming is larger for the transient initialisation simulations than for the equilibrium initialization simulations. Positive feedbacks drive the collapse of the WAIS, and these feedbacks are initiated earlier if the simulations start out-of-balance than in equilibrium. After the collapse of TG and PIG, the impact of ocean warming ceases in the transient initializations simulations, as the GMSL rise contribution slows down in the final phase of the collapse (Fig. 3b) and the WAIS can only collapse once. This ceasing effect will not occur for the equilibruim initialization simulations, as the
ocean forcing induces all the GSML rise contribution and the collapse of the WAIS, which otherwise would not happen.

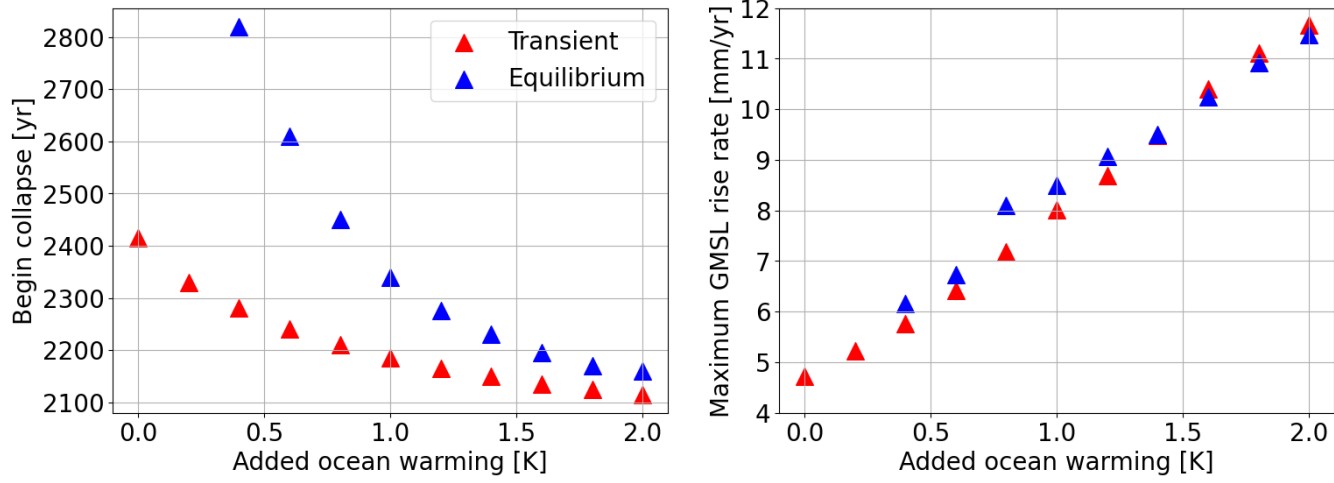

**Figure 5. Schematic ocean warming correlated to the beginning of the collapse and the maximum GMSL rise rate** (left) The beginning of the WAIS collapse defined as the ungrounding of ridge AB in Van Den Akker et al. (2025b) as function of additional uniform and sudden ocean warming in the ASE. Blue dots represent simulations. (right) the maximum GMSL rise rate modelled in the same simulations, also as
function of the added ocean warming.

In Figure 5, increasing ocean warming is correlated with an earlier onset of TG collapse, which in our simulations consistently acts as the precursor to WAIS collapse. In this study, all collapse events initiate at TG. The onset of collapse is defined as the first timestep at which the bedrock ridge, 50 km upstream the current grounding line (see Figure S6) becomes entirely free of
grounded ice. The role of this bedrock ridge is discussed in more detail by Van Den Akker et al. (2025b). The results show an asymptotic trend: higher ocean temperatures lead to earlier collapse, but with progressively smaller shifts in timing. This suggests the presence of an ice-dynamical limit, TG cannot collapse before ~2100 without more than 2 degrees of ocean



warming. Meanwhile, the maximum GMSL rise rate during the collapse increases linearly with ocean temperature, indicating that while earlier collapse timing shows diminishing returns, the rate of sea-level rise keeps on intensifying with stronger

forcing. Furthermore, simulations starting from an ice sheet in equilibrium have the collapse delayed by multiple centuries compared to the transient initialization simulations. Here, however, warmer ocean waters still bring the collapse onset forward, without approaching a limit. It is noteworthy that the maximum GSML rise rate, if reached, is comparable to transient initialization runs.

## 3.2 Realistically forced Future Antarctic Mass change

Figure 6 shows the simulated integrated ice sheet mass loss in the Amundsen Sea Embayment (ASE) as well as for the entire Antarctic Ice Sheet. Until 2150, the projected mass loss is largely determined by the initialisation method; hence the current dynamic imbalance sets the `short term` mass loss of the ASE. After 2150, all simulations project a collapse of the WAIS, most simulations by 2500, otherwise before 2700 (not shown). Simulations initialized with the transient initialization

configuration (red solid lines) predict a faster ice mass loss and an earlier collapse than those initialized with the equilibrium initialization configuration (blue dashed lines). This accelerated ASE collapse due to transient initialization occurs 50 to 100 years earlier under low emission scenarios (SSP126 or RCP26) and 25 to 50 years earlier under high-emission scenarios. These findings highlight the impact of the present-day imbalance on WAIS projections and emphasize the need to include those in regional simulations. The influence of this imbalance is more pronounced under cooler scenarios, where future ocean warming

and surface melting is less dominant. In warmer scenarios (RCP8.5 and SSP5-85), the strong ocean forcing and net negative SMB diminishes the relative impact of model initialization choices. The forcing then rules the response. Similar results have been described for glacial isostatic adjustment applications by Van Calcar et al. (2024) who argue that the details of the GIA matter relatively more for small forcing and relatively less for strong forcing.

For the whole AIS and in the perspective of modelled multi-metre GMSL rise contribution, the present-day imbalance has an impact on the AIS mass loss before 2100. All simulations starting from the transient initializations show more integrated mass loss over the AIS in 2100 compared to the simulations starting from the equilibrium initialization, with in most cases a doubled GMSL rise contribution when using the present-day observed mass change rates, except for the simulations forced by HADGEM_RCP585. This forcing contains early ocean warming and little increases in the SMB in the beginning of the

simulation, overshadowing almost immediately the differences caused by the transient and equilibrium initialization with a much larger signal.

After 2150, in our simulations the emission scenario has the largest impact on the projected mass loss, for the high-end and low-end scenarios the projected GMSL rise is $1 - 6$ and $1 - 2$ m, respectively. The difference in projected ocean changes by

various ESMs is the next largest source of uncertainty, this will be discussed more below, Lastly, the effect of equilibrium or





transient initialisations on the projected GMSL rise from the AIS is, in relative sense, profoundly smaller than in regional simulations focussing on the ASE, or on short term projections of the AIS.

For high-emission scenarios, incorporating the mass change rates even decreases the GMSL rise contribution. This can be
explained by the strength of the warming in these scenarios, which leads to the collapse of major ice shelves in both transient and equilibrium initialized simulations. At the same time, observed thickening in parts of the grounded East Antarctic Ice Sheet (EAIS) continues in the future simulations, offsetting some of the mass loss from West Antarctica. As a result, mass loss in the WAIS is partially balanced by the increase in thickness in Dronning Maud Land, yielding a similar projected GMSL rise contribution, see Figure 6. We argue that the spatial distribution of mass change in the transient initialization simulations
is more physically justified compared to the pattern of mass loss in the equilibrium initialization cases, since the former captures a projected thickening of the EAIS until 2100, in line with recent studies on the forced future of the modelled Antarctic Ice Sheet (Siahaan et al., 2022; Coulon et al., 2024; Klose et al., 2024; O'neill et al., 2025) and in line with present-day observations (Smith et al., 2020).

Simulations forced with output from NorESM (RCP126 and RCP585) are outliers. Notably, the simulation NorESM-RCP126 is the only one that projects less ice mass loss in the ASE region compared to the estimated mass loss from the present-day dynamic changes in the absence of any future forcing (black lines in Fig. 6). Additionally, the simulation applying NorESM-RCP585 leads to AIS mass losses comparable to mass losses originating from the simulation using UKESM-SSP126, likely because NorESM is run towards 2100, and then the forcing is repeated until 2300, rather than running with an increase in
ocean forcing until 2300. In other words, the warming in NorESM-RCP585 triggers the same kind of modelled ice sheet response as the warming in UKESM-SSP126. NorESM's relatively low climate sensitivity of 2.5 K also helps explain this behaviour (Seland et al., 2020).

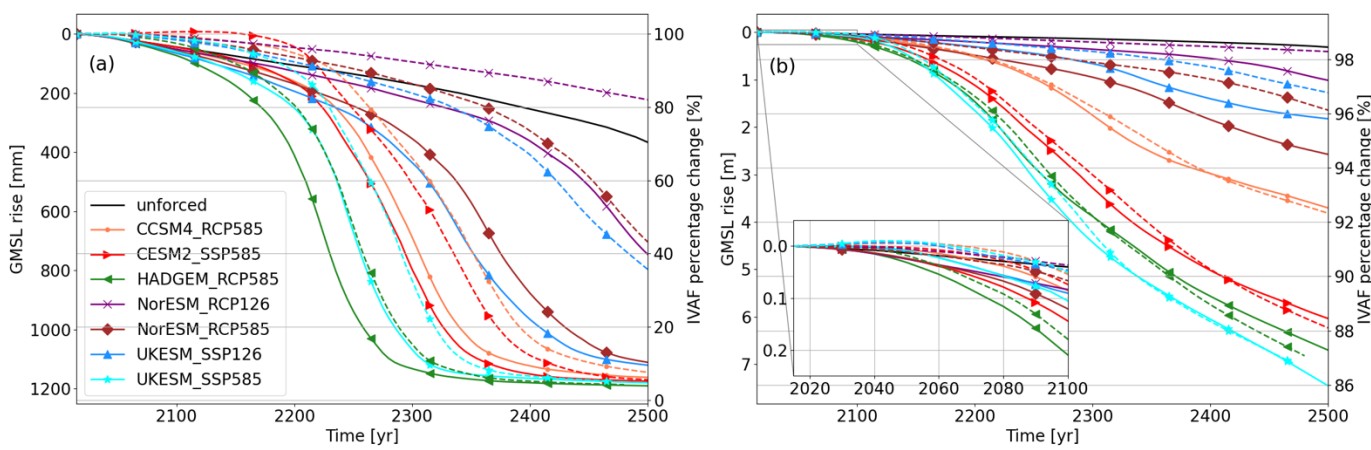






**Figure 6.** Integrated mass loss in the ASE, where Thwaites Glacier and Pine Island Glacier are situated (a). Left y-axis shows the GMSL rise in mm and right y-axis the Ice Volume Above Floatation in percentage of the initial value. On the right, the same for the whole AIS (b). Solid lines indicate simulations starting with the observed mass change rates, dashed lines indicate simulations starting from an equilibrium. Colors indicate the ocean forcing scenario used. The black line indicates an unforced transient initialization simulation.


Next, we discuss the modelled patterns of ice shelf and grounding line retreat. Figure 7 shows the number of simulations containing ice (Fig. 7a) and grounded ice (Fig. 7b) across Antarctica for the year 2500. The most significant reduction in grounded ice area occurs in WAIS. Aside from Wilkes Land, the grounding line retreat is limited in the EAIS as for most of the coastline, the inland bedrock is close to or above sea level. All simulations predict substantial grounded ice loss in Thwaites

Glacier, as already clear from Figure 6a. While not all scenarios result in a full WAIS collapse, several high-warming scenarios show complete loss of floating ice at the current locations of PIG and TG.

For half the simulations, the Filchner Ronne shelf and the Bungenstock Ice Rise (near the Southern margin of the Ronne ice shelf) have respectively disappeared and completely ungrounded entirely by 2500. Even more than half (9 out of 14)

simulations end with the Ross shelf completely melted, in some cases along with glaciers at the Siple coast. As our simulations does not model calving for retreated ice shelves, this disintegration is entirely due to enhanced basal melting by warmer ocean waters. In fact, simulations where the Filchner-Ronne or the Ross shelves disintegrate, the Bungenstock Ice Rise or the Siple Coast deglaciate, showing a direct relation between the large floating ice shelves and their upstream tributary glaciers.

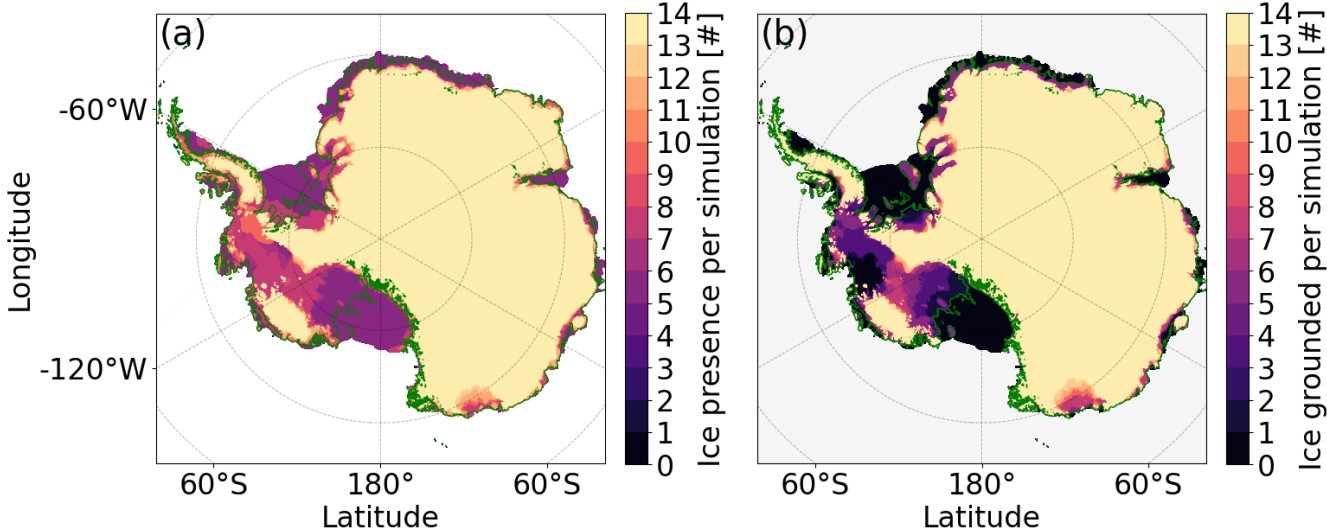

**Figure 7.** Ice present (a) and grounded (b) at the end (2500) of every simulation per grid cell, summed over all 14 simulations: with 7 ESM scenarios and the 'transient and equilibrium initialization.



This clear connection between floating and grounded ice mass is further assessed in Figure 8. The floating ice mass and percent change of IVAF since 2015 are shown per region (Filchner-Ronne, Ross and ASE as delineated in Fig. S3) in this figure. Both

the Filchner-Ronne and Ross shelves display similar trends. Some ESM forcings result in little change in modelled floating ice mass on the shelves during the simulation. For the Ross shelf, this occurs under UKESM-SSP126 and NorESM_RCP26, while for Filchner-Ronne the same applies with the addition of NorESM_RCP85. In all other scenarios except NorESM_RCP85 for Ross, the volume of floating ice diminishes. This pattern is irrespective of the transient or equilibrium initialization: the dashed and solid lines in Figure 8 show little difference. These regions are currently relatively stable

compared to the ASE (Smith et al., 2020). As a result, incorporating the small observed mass change rates in those regions does not affect the mass loss considerably in future projections.

The loss of floating ice mass is closely tied to the average thermal forcing in each region, as shown in Figure 8. For the Filchner-Ronne (FR) shelf, the percent change in IVAF, the floating ice mass loss and the average thermal forcing all follow

a consistent pattern. Some ESMs simulate increased thermal forcing at the FR calving front, which leads to near-complete shelf collapse and substantial ice mass loss. Others maintain thermal forcing near present-day levels, resulting in a largely intact ice shelf and minimal mass loss. A similar pattern is observed for the Ross shelf, the middle column in Figure 8. The notable exception is NorESM_RCP85, which leads to a reduced Ross shelf in 2300 but not to a complete loss. In this simulation, thermal forcing rises sharply between 2050 and 2100 before stabilizing, which results from repeated climate forcing

applied to the NorESM simulations. The early spike is enough to halve the shelf volume quickly, the subsequent stabilization allows the remaining half to persist.

The ASE region exhibits a distinct pattern. At the beginning of each simulation, there is little floating ice present in the PIG and TG basins. The present-day shelves are small compared to FR and Ross. As soon as the grounding line starts to retreat and

the grounding line flux increases, a large shelf is formed. However, if the thermal forcing is strong enough, this newly formed shelf rapidly melts, producing the increasing and then decreasing pattern of floating ice mass loss in the bottom row of Figure 8. Van Den Akker et al. (2025a) provide a more detailed discussion of the role of the shelves on the future dynamics of TG.





**Figure 8. IVAF, floating ice mass and average thermal forcing per basin until 2300.** The ice volume above floatation as percentage of what was present at the initialization (left column), the total mass of floating ice per section (middle column) and the average thermal forcing at 510 m below sea level at the calving fronts for Ross (upper row), Filchner Ronne (middle row) and ASE (bottom row). Forcing is shown until 2300, after 2300 it is kept constant.

The loss of grounded ice mass and the reduction in floating ice volume are similarly related for the Ross and Filchner-Ronne basins. Once a critical fraction of floating ice is lost, the buttressing declines, leading to accelerated upstream flow of grounded ice, contributing to GMSL rise. To estimate this time lag between the onset of IVAF reduction and grounded ice loss, we calculated the time delay between 10% IVAF loss of an ice shelf sector and 100 mm of GMSL rise from that sector for each simulation and for the Filchner-Ronne and Ross sectors. We excluded the ASE from this analysis as in this region buttressing has currently little impact on the ice dynamics (Favier et al., 2014; Robel et al., 2019; Lipscomb et al., 2021; Gudmundsson et al., 2023; Van Den Akker et al., 2025b). Typically, the delay between losing 10% of floating ice and reaching 100 mm of GMSLR contribution is in the order of decades (Table 4). In general, glaciers feeding into the Filchner-Ronne Ice Shelf respond slightly more rapidly than those along the Siple Coast, indicating that the Filchner-Ronne shelf provides a little more





buttressing. Therefore, once 10% of the Filchner-Ronne's floating ice volume is lost compared to present-day levels, most
simulations show that 100 mm of GMSL rise follows within roughly a century.

Typically, except for NorESM_RCP85, the delay in deglaciation of the FRIS is higher when doing a transient initialization
compared to simulations starting from the equilibrium simulation, while the delay for Ross is often shorter. This is due to an
opposite pattern in the mass change rates of Smith et al. (2020). The Filchner-Ronne shelf is currently thinning, but the
grounded ice upstream is slowly thickening. This is reversed at the Siple Coast and the Ross ice shelf: here the shelf is
thickening and the grounded ice mainly thinning. Adding the mass change rates as was done for the transient initialization
makes the grounded ice upstream of the Filchner-Ronne shelf slightly more stable and at the Siple Coast slightly less stable,
and therefore slightly more sensitive to floating ice mass loss.


**Table 4. Delay in years between a floating ice mass loss of 10% and 100 mm SLR contribution.** The symbol 'x' denotes
simulations when the 10% ice volume above floatation threshold and/or the 100 mm SLR contribution is never reached. The
Filchner-Ronne shelf is abbreviated with 'FRIS', the Ross shelf with 'Ross'.

| ESM forcing | Transient initialization delay (yr) | | Equilibrium initialization delay (yr) | |
|---|---|---|---|---|
| | **FRIS** | **Ross** | **FRIS** | **Ross** |
| CCSM4_RCP85 | 70 | 150 | 65 | 155 |
| CESM2_SSP585 | 70 | 75 | 65 | 85 |
| HADGEM_RCP85 | 60 | 80 | 55 | 85 |
| NorESM_RCP26 | x | x | x | x |
| NorESM_RCP85 | 160 | 110 | 185 | 110 |
| UKESM_SSP126 | x | x | x | x |
| UKESM_SSP585 | 55 | 40 | 40 | 40 |


Finally, in Figure 9, we test again our null hypothesis by subtracting the no-forcing experiment from all forced experiments.
Compared to Figure 4, the differences between the transient and equilibrium simulations are much less pronounced, hence the
long term impact of the current imbalance and future warming looks to add up largely linearly. Only when the transiently
initialized simulation is in the collapse phase of TG and PIG, the ESM forcing causes a non-linear enhencement of the GMSL



rise contribution from the ASE sector (Fig. 9a). This is caused by enhanced ocean warming in the ASE sector compared to the initializations, which is most pronounced in the HADGEM_RCP85 scenarios. The other simulations shown in Fig 9 that show a distinct difference in response between the equilibrium and transient simulations are UKESM_SSP585 and CESM2_SSP585, both of which include severe drops in the surface mass balance over the grounding line are of the ASE sector (see Fig 2.).

All other ESM forcing datasets contain a smaller difference in ocean temperature with the spinup (see Fig 1.), as well as smaller SMB anomalies, and therefore show a less pronounced difference with their respective control scenario.

Prior to the collapse phase, the simulations align much more closely than in Figure 4. This may be explained by the fact that, as the TG and PIG basins approach collapse, their mass loss becomes increasingly governed by ice dynamics and less by

external forcings. Consequently, simulations starting from equilibrium conditions are more sensitive to gradual warming, which corresponds to the faster increase in GMSL contribution observed in the transiently initialized simulations (Fig. 8, bottom row). As in Figure 4a, the influence of the ESM forcing ceases after the collapse, since WAIS also collapses in the transient simulation without forcing—demonstrating that WAIS can only collapse once. Lastly, the difference between the transiently and equilibrium-initialized simulations is minimal for the AIS as a whole (Fig. 9b), as the substantial projected

GMSL contribution by 2500 from AIS primarily originates from regions outside the ASE sector, which currently remain in balance (Fig. 8).

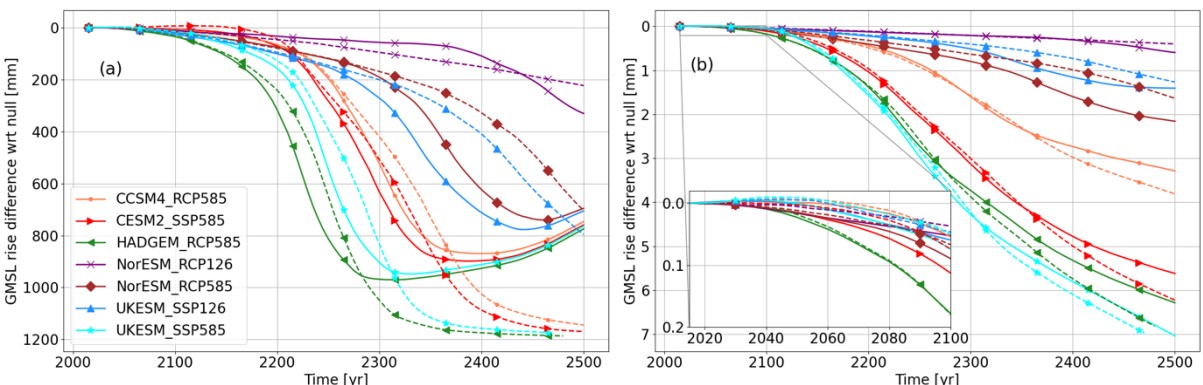

**Figure 9. Extra induced mass loss by the ESM forcings with respect to the unforced simulations**, for (a) the ASE and (b) the AIS in

terms of GMSL rise. Solid lines indicate simulations starting with transient initialisation, dashed lines indicate simulations starting from the equilibrium initilisation. Colors indicate the ocean forcing scenario used.

## 4. Discussion

In this study, we present that including the observed present day mass change rates in an ice sheet model (CISM) improves the quality in projected ice mass loss for the coming century (up until 2100), because it is consistent with currently observed





GMSL rise contribution. Before 2100, including the present-day mass change rates lead to considerably higher GMSL rise contributions from the AIS, regardless of the ESM forcing chosen. After 2100, dynamic effects like a TG and PIG collapse start to develop, leading to accelerating mass loss. Including the present-day mass change rates accelerates a modelled WAIS
collapse by 25 to 100 years in forced simulations.

Our schematic experiments show that simulations initialized with present-day mass change rates respond less strongly to schematic and uniform ocean warming than those using equilibrium initializations. As expected, greater ocean warming triggers an earlier collapse of TG and, subsequently, the ASE basin but with diminishing returns: the most significant impact
of additional warming is seen in the low-warming scenarios. Notably, the onset of collapse does not occur before 2100, even with 2 degrees of ocean warming. This suggests a potential ice dynamical threshold that delays the TG collapse, indicating that extreme warming may not accelerate the collapse itself significantly. However, the peak rate of GMSL rise during the collapse continues to increase linearly with additional ocean warming. This alligns with the contrasting patterns in Figures 4 and 9: while the non-linear enhancement of mass loss by additional ocean warming and present-day mass losses prior the
collapse of the ASE sector is very apparent in the idealized simulations, it is largely absent in the simulations forced with ESM data. Hence, our null hypothesis, that the present-day mass loss rates do not influence future forced projections, is effectively incorrect, but the non-linear behavior willl probably not materialize and is only potentially relevant as long as the ASE sector is the primary contributor of the AIS to GMSL rise.

We argue that including the present-day observed mass change rates is necessary for regional TG and PIG ice model simulations, and for shorter (until 2200) simulations for the entire AIS. The present-day imbalance is more relevant for simulations forced with lower climate forcing, as in simulations with RCP8.5 and SSP5-8.5, the climate forcing dominates the ice sheet response on the long term (2200 – 2500). In those cases, the impact of specific choices made during the initialization procedure are becoming negligible compared to the impact of the large increases in ocean temperatures yielding a rapid
collapse of the ice shelves.  Especially on the continental AIS on long time scales, adding the mass change rates to the initialization has limited impact on the projected sea level rise until 2500. Here, the choice of ESM forcing provides more variability. This is mainly caused by the ocean temperatures increases, which causes the loss of the large ice shelves. Some forcing scenarios cause the Filchner-Ronne and Ross shelves to disappear. These areas, and their upstream grounded tributary glaciers, are at present-day not exhibiting large mass change rates, and including the present-day mass change rates did
therefore not change the initialized ice sheet much.

A clear connection between ocean thermal forcing and floating, and consequently grounded, ice loss was found: significant floating ice volume loss leads in most cases within a century to significant grounded ice loss and sea level rise. This can be explained by two mechanisms: the sensitivity of the basal melt parameterization to changes in the ocean temperatures (e.g. the
sum of $TF_{\mathrm{base}}$ and $\delta T$ in Eq. (1.5)) and the location of large SMB anomalies. Regarding the former, when deriving Eq. (1.5)



with respect to the sum of $TF_{\mathrm{base}}$ and $\delta T$ and filling in parameter values, we find that the basal melt parameterization has a temperature change sensitivity of ~11 m yr$^{-1}$ K$^{-1}$ when the sum of $TF_{\mathrm{base}}$ and $\delta T$ is 1 K, increasing linearly (e.g., when the sum is 2 K, the melt sensitivity is 22 m yr$^{-1}$ K$^{-1}$). For some scenarios, the ocean forcing applied can be several degrees K, causing an increase in basal melt of hundreds of metres per year, while the SMB anomalies range only from -2 to 2 m yr$^{-1}$.


The relative unimportance of the SMB changes is related to the location of their anomalies. The largest anomalies are found near the coast of the AIS, where in many locations floating ice, or as the simulations progress, no ice at all exist. Therefore, the increased/decreased SMB is not contributing significantly to changes in projected GMSL rise from simulations forced by output from different ESMs. However, the SMB in the CMIP5 and CMIP6 models are determined using the present-day

geometry of the Antarctic Ice Sheet. In our simulations, the ice sheet geometry changes drastically, with surface heights decreasing for most of the WAIS. Lower surfaces will through the lapse rate yield higher air temperatures and a more negative SMB, which could enhance mass loss more when this effect is incorporated. Ideally, ice sheet modellers should use coupled simulations where atmospheric models directly resolve the SMB of ice sheets based on their evolving geometries. This is computationally expensive and only done for the AIS in recent UKESM studies, such as shown by Siahaan et al. (2022).

Alternatively simple parameterizations of this effect could be used (e.g. Fortuin and Oerlemans (1990). A solution could be to use SMB emulators, but these have yet to be developed.

At the start of our simulations, we find a sea level rise rate in the order 0.1 – 0.5 mm per year, when using the transient initialization method. This is in line with the observed rates reported by Cronin (2012); Smith et al. (2020); Fox-Kemper et al.

(2021). In 2100, our spread in projected sea level rise from the WAIS and AIS is about 5 – 25 mm, which is comparable to the present-day observed rate (approximately 0.3 mm yr$^{-1}$) and in line with Van De Wal et al. (2022) which . This range is similar to values reported by Edwards et al. (2021); Coulon et al. (2024); Klose et al. (2024); Seroussi et al. (2023); O'neill et al. (2025). In our ensemble, we do not find any cases where the AIS gains net mass during our simulated period, contrasting results found by Siahaan et al. (2022). They found increased snowfall to dominate over increased basal melting, which leads

to a net mass gain until 2100, with higher mass gains with warmer climates. However, in simulations done with low climate forcing, they found a steady decrease of ice mass at the WAIS, similar to the results presented in this study. When continued, this could in their simulation lead to the Marine Ice Sheet Instability and further enhanced mass loss beyond 2100, possibly outpacing mass gain through increased snowfall.

In 2300, our modelled ensemble shows a GMSL rise contribution of roughly 100 – 1200 mm from the WAIS and 100 – 4500 mm for the entire Antarctic ice sheet. This large range is caused by dynamic instabilities caused by ice sheet thinning earlier in the simulation. Our ensemble almost captures the range reported by Seroussi et al. (2024); Greve et al. (2023); Payne et al. (2021), with the exception of the cases where the AIS is growing (i.e. when a negative GMSL rise contribution is simulated). The dynamic instability leading to the WAIS collapse is featured in all our simulations, causing large mass losses and thereby



compensating any (small) mass gains on the EAIS. Furthermore, our ESM forcing used shows predominantly warming ocean
waters and a decreasing SMB with time, in contrast to Siahaan et al. (2022), in whose simulations till 2100 the SMB increase
dominates over ice dynamical processes.

In many simulations shown in our study, all floating ice disappears from the Filchner-Ronne and Ross shelves. This is not
uncommon in forced AIS simulations (Coulon et al., 2024; Seroussi, 2021; Seroussi et al., 2024). However, the disappearance
of the big ice shelves is controlled by the amount of warming available in their cavities. CISM does not contain a submodule
to simulate the overturning circulation in the ice shelf cavities, and neither did the ESMs from which the forcing was used in
this study. The thermal forcing was only available in the open ocean bounded by the (observed) calving front of the AIS, and
had to be extrapolated into the cavities. Therefore, it cannot be ruled out that the warming simulated in the cavities that lead
to dramatic floating ice loss of the large ice shelves is caused (at least in part) by the extrapolation scheme. In essence the
extrapolation scheme now determines when the ice shelf cavities shift from a cold cavity, which they are at present, to a warm
cavity. Future studies could use cavity-resolving ocean models (Scott et al., 2023), or an intermediate complexity 2D layer
resolving model like LADDIE (Lambert et al., 2022).

Simulations forced by the output of NorESM are outliers. NorESM for both scenarios project little warming in the Southern
Ocean and no large change in the SMB (patterns) compared to present-day, because it is only dependent on repeated forcing
from 2100 rather than increased climate change until 2300. Furthermore, the cooler trend could be caused by the formation of
gyres just outside of the Ross calving front in NorESM simulations (Seland et al., 2020), which cool the ocean just outside of
the continental margin. Interpolating ocean data from just outside the continental margin or the calving front will then lead to
a cold bias. Furthermore, NorESM is known to have a low climate sensitivity, and it can take several centuries for it to warm
as much or more than for example CEMS2 or UKESM.

In this study's simulations, we applied a no-advance calving approach: the calving front was restricted from advancing beyond
its present-day observed position. While it was allowed to retreat, primarily due to significant increases in basal melt rates
observed in most simulations, calving ceased entirely once the front retreated upstream of its current position. Although this
is a conservative and somewhat unphysical representation, it serves as a simplified framework. Incorporating a more physically
grounded calving law, accounting for factors such as stress and strain rates, rather than relying solely on modeled front
positions, could improve predictions of ice mass loss. Integrating such advanced calving schemes into CISM is an area of
active research.


## 5. Conclusion

In this paper, we show ocean-forced Antarctic wide simulations conducted by the Community Ice Sheet Model up until 2500.
We test a new feature of the model: initializing with the present-day observed mass change rates. Schematic ocean warming





causes a faster onset of TG collapse with diminishing returns but linearly increasing GMSL rise rates with additional warming.
We furthermore find that including the present-day imbalance is important for regional WAIS simulations, where including
the present-day mass change rates in a simulation speeds up Thwaites Glacier and Pine Island Glacier collapse by 25 – 100
years. Including the mass change rates doubles the AIS GMSL contribution in 2100 in our ensemble. For long-term continental
AIS simulations beyond 2100, the choice of the ESM used is more important than the choice whether or not include the present-
day mass change rates in a century timescale simulation. We find for all simulations that the AIS will continue to lose mass
over the next five centuries, with uncertainties increasing strongly with time.

Our study highlights that, for simulations until 2500, the main ice losses happen in the Ross and Filchner-Ronne, preceded by
mass loss in the ASE region, but the pattern is highly depedent on the extrapolation scheme of ocean properties into the ice
shelf cavities. The mass balance of the floating ice shelves proves to be crucial for the grounded ice loss rate. In our simulations
we do not employ a physically based calving scheme or a sub-shelf cavity resolving ocean model. Replacing both processes
with physically based parameterizations or sub-models will likely change the mass balance of the floating shelfs and ultimately
increase our confidence in sea level rise projections from the Antarctic Ice Sheet. Including a more physically based rather
than a location based calving flux will probably increase the modelled future mass loss of the AIS, and therefore project faster
ice sheet retreat and as a consequence, more GMSL rise.


**Code availability**

CISM is an open-source code developed on the Earth System Community Model Portal (EPSCOMB) Git repository available
at https://github.com/ESCOMP/CISM. The specific version used to run these experiments will be tagged before publication.

**Data availability**

The output of the simulations shown in this study will be uploaded to a Zenodo repository before publication.

**Author contributions**

TvdA designed and executed the main experiments. WHL and GRL developed CISM and helped configure the model for the
experiments. RSWvdW, WJvdB, WHL, GRL provided guidance and feedback. TvdA prepared the manuscript, with
contributions from all authors.

TvdA received funding from the NPP programme of NWO. WHL and GRL were supported by the NSF National Center for
Atmospheric Research, which is a major facility sponsored by the National Science Foundation (NSF) under Cooperative
Agreement no. 1852977. Computing and data storage resources for CISM simulations, including the Derecho supercomputer
(https://doi.org/10.5065/D6RX99HX), were provided by the Computational and Information Systems Laboratory (CISL) at
NSF NCAR. GRL received additional support from NSF grant no. 2045075.



The authors declare no competing interests.

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
