# Peer review of "The effect of the present-day imbalance on schematic and climate forced simulations of the West Antarctic Ice Sheet collapse"

_EGUsphere, 2025_

## Author Comment (AC1)

**Response to reviewers: The effect of the present-day imbalance on schematic and climate forced simulations of the West Antarctic Ice Sheet collapse**

We would like to start with a general expression of gratitude to the reviewers for their positive attitude, constructive feedback and helpful suggestions to improve the manuscript. Please find our replies in blue.

**Referee #1 (anonymous)**

The paper presents a series of model experiments on the evolution of Thwaites Glacier over the next centuries using an ice sheet model forced by several climate models. The paper demonstrates that the historical imbalance of the glacier matters a lot for its future stability and its potential for 'collapse'. The authors also put forward a limit in global temperature increase for which the glacier could 'collapse'.

The paper is rather lengthy and could benefit from some trimming, which would make the message clearer. Especially the experiments of steady state versus transient initialization are of interest, followed by the forcing experiments. The introduction on the different ways of initializing models could be shortened, as the importance for the paper is to make the distinction between steady state and including imbalance.

Overall, I find this an interesting study that with some polishing and a few clarifications (see below) I would recommend for publication.

We thank the referee for their positive review and constructive comments that will improve the manuscript, and will discuss below how we implemented these comments.

Line 14: model -> models

Adopted

Line 14: what models are meant here. I guess climate models and not ice sheet models. Please specify.

We meant ice sheet models, forced by climate models. We clarify this in the revised manuscript, by rephrasing this sentence to: "....".

Line 19: Collapse occurs 58 times in the text, but it is never defined what is exactly meant by collapse of the ice sheet. Later on 'onset of collapse' is used, which also requires a clarification.

By "collapse" we mean accelerated deglaciation leading to considerable grounded ice mass loss. We identify the onset of collapse as the first timestep at which the bedrock ridge 50 km upstream of the current grounding line (as shown in Figure S6) becomes entirely free of grounded ice. This is a convenient definition since the deglaciation accelerates once the ice is free of this pinning point. We will clearly define these terms in the revised manuscript as: collapse (i.e. accelerated deglaciation leading to considerable grounded ice mass loss)

The introduction is quite long giving a complete overview of different methods of initialization. It is quite interesting in itself, but is not necessarily guiding the reader towards

the core of the paper, i.e., that starting a historical simulation from an observed imbalance results in different response of TG compared to starting from steady-state conditions. It is not so much the way an initialization is done, but what the imbalance is that counts for understanding the remaining of the manuscript.

We will shorten  the introduction and focus on the forced collapse of the WAIS with and without the present-day imbalance.

Line 91: Our null hypothesis is that the GMSL rise from the present-day mass loss rates is independent of the GMSL rise caused by an increase in ocean thermal forcing, i.e. that the present-day mass loss rates do not influence future forced projections.

Quite confusing. I would suggest to remove the mention to GMSL. It is about mass loss either caused by a given imbalance due to a grounding line retreat some time ago, or due to the current applied ocean forcing. I don't see how GMSL rise can be caused from thermal forcing (except thermal expansion, but that is not what you are talking about I presume).

We will rewrite these lines, also in reply to referee #2, to: 'With these simulations we will investigate the importance for the future AIS evolution of the current mass imbalance,compared to that of future changes in ocean temperature and SMB. Our null hypothesis is that incorporating the present-day imbalance does not influence the forced deglaciation of the WAIS.'

Line 100: Is a spatial resolution of 4km sufficient to guarantee grounding line migration (see for instance discussion in Pattyn et al., 2013). Maybe briefly state what is done to facilitate grounding line migration at such spatial resolution.

Yes, we think 4km resolution is sufficient to capture grounding line dynamics, especially with the grounding line parameterization of Leguy et al (2021).

'We run CISM on a uniform 4 km grid,  justified below. using the grounding line parameterization from Leguy et al. (2021) which scales the basal sliding and melt rate proportionally to its grounded and floating area fraction respectively. Doing so, Leguy et al. (2021) showed that this resolution is adequate to capture grounding line dynamics using CISM and idealized marine ice sheet experiments, Lipscomb et al. (2021) showed that this resolution in combination with the scaling reduces the model result's grid resolution dependency when modelling the AIS.'

Line 109: The regularized Coulomb friction law was already used in Joughin et al (2019), and is based on the work of Schoof and Gagliardini. (Joughin, I., Smith, B. E., and Schoof, C. G.: Regularized Coulomb Friction Laws for Ice Sheet Sliding: Application to Pine Island Glacier, Antarctica, Geophys. Res. Lett., 46, 4764–4771, https://doi.org/10.1029/2019gl082526, 2019.)

We will rewrite line 109 as: 'We use the regularized Coulomb sliding law suggested by Joughin et al. (2019) and confirmed with laboratory experiments by Zoet and Iverson (2020)'

Line 121: The reference that marine sediments are likely more prevalent in submarine basins may be a bit outdated. There are more recent studies that have investigated the probability of sediment versus hard bed of Antarctica. See for instance:

https://agupubs.onlinelibrary.wiley.com/doi/full/10.1029/2021RG000767 and
https://www.nature.com/articles/s41561-022-00992-5

It shows a larger diversity of possible outcomes for regions lying below sea level.

These are good suggestions and references, and challenge the first-order approximation of Aschwanden et al (2013). We will  add after line 121: 'The parameterization and associated values used to represent marine sediments in Aschwanden et al. (2013), which rely solely on bedrock elevation, are challenged by the more recent findings of Li et al. (2022), who show that the likelihood of marine sediment does not directly correlate with bedrock height. Incorporating the likelihood map of  Li et al. (2022) into CISM and analysing the influence of marine sediments on marine-based ice sheets is beyond the scope of this study, but represents a promising direction for future work.'

Line 127: What are these parameters (Ho, tau, r L)? How do they influence your optimization? It is not defined what these parameters are about.

Thank you for pointing this out. We revise the manuscript in the following manner to explain these parameters:: 'In 1.4a, $H_0, \tau, r$ and $L$ are scaling constants, used to adjust the relative weights of the different terms. Increasing/decreasing $H_0$ makes the optimalization less/more sensitive to ice thickness errors; increasing/decreasing $\tau$ makes the changes in $C_l$ per timestep smaller/larger, increasing/decreasing $r$ draws $C_l$ more/less to the relaxation target $C_{lr}$; and increasing/decreasing $L$ results in a smoother/spikier 2D pattern of optimized $C_l$. Their values were tested and chosen to represent AIS thickness and surface velocities well, with minimal drift. Table S2 shows their values.'

Line 158: See my remark of Line 100: is this the way grounding line migration is dealt with? Interpolation of friction within partially floating cells AND subshelf melt as well? It has been shown in Seroussi and Morlighem (https://tc.copernicus.org/articles/12/3085/2018/)  that it increases the sensitivity of grounding line retreat big time. Some discussion is needed.

Yes, friction and subshelf melt in grounding-line grid cells are scaled to their respective grounded and floating area fraction. This choice is based on the results of Leguy et al. (2021), who  showed that: (i) when using this so-called partial melt parameterization (PMP), the resolution dependency of grounding line movement in CISM is the lowest, (ii) grounding line parameterization sensitivity is model dependent, so what works for ISSM doesn't necessarily work for CISM, (iii)  using a comparable simulation setup as in Seroussi and Morlighem (2018), results using a resolution of 4km capture grounding line migration accurately–  and are comparable to those with a resolution of 2km and coarser. We agree with the referee that this point is not made clearly in the manuscript, and thank the referee for suggesting this reference. We will add to line 170:

"Schematic tests by Seroussi and Morlighem (2018) showed that applying basal melt in proportion to the floating area fraction can lead to an overestimation of grounding line retreat rates; they therefore discouraged the use of PMPs. However, Leguy et al. (2021) conducted similar schematic tests with CISM and found that using a PMP reduced the CISM's sensitivity to grid resolution more than the No-Melt Parameterization (NMP) recommended by Seroussi

and Morlighem. In more realistic AIS applications of CISM, Lipscomb et al. (2021) found that the PMP produced a moderate sensitivity of grounding line migration rates to grid resolution, lower than the sensitivities to basal melt rate and basal friction parameterizations. These results suggest that the optimal GLP is model-dependent and that for CISM, 4-km grid resolution using a PMP is sufficient for modeling continental-scale ice sheets on multi-century timescales."

Line 165: isn't this not too much different than keeping the calving front fixed, as you probably need quite high melt rates to have the front retreating through melting alone.

Yes, it is. We will add 'In practice, this means that the calving front will retreat only when the basal melt rates are increased greatly, and for present-day conditions, the calving front position is fixed at its observed location'.

Line 181: Is an initialization of 10 ka enough for the temperature field to reach an equilibrium?

We start an initialization with the Robin solution to the temperature profile. During this 10 ka, the AIS geometry does not change considerably (with the exception of some outlet glaciers, mainly in the peninsula). The average temperature profile of grounded, floating and all modelled ice grid cells is shown below, which we will add to the supplementary material:

[Figure]

**Figure S3. Grid point average temperature during the transient initialization (similar to the equilibrium initialization, not shown)** for floating ice (red line, right y-axis), grounded ice (blue line, left y-axis) and all grid points containing ice (black line, right y-axis)

We will also add to line 189: Figure S3 shows the evolution of the average temperature during the initialization: it flattens out at 10 kyr, indicating that the ice sheet has reached thermal equilibrium.

Table 1:  Overall, I found the figures in the supplementary material more of interest than the first few figures shown in the manuscript. Therefore, some information of figures S4 and S5 could be transferred to the main manuscript and replace table 1. One way of representing this is as in Martin et al, (2011) Figure 15 (https://tc.copernicus.org/articles/5/727/2011/tc-5-727-2011.pdf), so that different regions of the ice sheet/ice shelf system are represented.

This is a good suggestion. We will add the following figures to the manuscript, and add the values shown in Table 1 to the caption of the figures:

[Figure]

Figure 3. Binned ice thickness (m) for the observed (solid) and modelled ice (dashed lines). The present-day condition of the transient initialization is shown on the left, the equilibrium simulation on the right. For the transient initialization, the root mean square errors (RMSEs) for floating ice, grounded ice and in total are respectively 44, 31 and 35 meters. For the equilibrium initialization they are respectively 50, 23 and 30 m.

[Figure]

Figure 4. Binned ice surface velocities (m yr$^{-1}$) for the observed (solid) and modelled ice (dashed lines). The present-day condition of the transient initialization is shown on the left, the equilibrium simulation on the right. For the transient initialization, the Root Mean Square Errors (RMSEs) for floating ice, grounded ice and in total are respectively 201, 112 and 143 m yr$^{-1}$. For the equilibrium initialization they are respectively 202, 98 and 130 m yr$^{-1}$.

We will rewrite  Section 3.1 accordingly.

Figure 3: Is not showing integrated mass loss, but mass contribution to SL in terms of GSLR and % of VAF. Mass loss also comprises that mass that is lying underneath floatation level.

That is correct. We will replace 'integrated mass loss' with 'Loss of ice volume above flotation' and check the whole manuscript where we now use "mass" if IVAF is a more appropriate quantity.

Line 307: I don't think that delta T can be considered an inverted parameter, as there is not inversion method used. Maybe use 'optimized'.

Yes you are correct. We will replace the words "inverted" and "inversion" by "optimized" and "optimization" respectively throughout the manuscript (we used similar wording after equation 1.4 (a,b) for example).

 Figure 4 and Line 349: instead of pointing the readers to a supplementary figure S6 just to find out where a little line is drawn, it would be more informative to mention in the caption where this line is situated as a function of present-day GL position (i.e., XX km inland from the current GL position). This line is also defined as bedrock ridge and important as onset of collapse. What is meant by onset of collapse (see also remark on collapse in general)?

This is a good suggestion. We added to the caption: 'Triangles indicate the timestep when the bedrock ridge approximately 50  km inland of the present-day TG grounding line ungrounds, depicted by the line AB in Fig S6'.

Line 400: I wouldn't call these simulations outliers. They are valid solutions for that given forcing. Just that these forcings are relatively low in melt and high in accumulation and therefore result in less mass loss than other forcings. This is not the definition of an outlier.

We agree.  As the other reviewer pointed out, it is relevant that NorESM was run  run until 2100, while the other four ESMs run till 2300.We will add the following text:

The simulations forced with NorESM output (RCP126 and RCP585) show much less mass loss than the others, as both NorESM simulations have less warming and more enhanced accumulation than the other ESM runs. The NorESM output assumes a constant climate after 2100, while output from the other ESMs assumes ongoing warming through 2300.